# SDOH-NLI: a Dataset for Inferring Social Determinants of Health from Clinical Notes

**Adam D. Lelkes**[1]*, **Eric Loreaux**[2]†, **Tal Schuster**[1], **Ming-Jun Chen**[1]‡, **Alvin Rajkomar**[1]‡

[1]Google Research, [2]Curai Health

## Abstract

Social and behavioral determinants of health (SDOH) play a significant role in shaping health outcomes, and extracting these determinants from clinical notes is a first step to help healthcare providers systematically identify opportunities to provide appropriate care and address disparities. Progress on using NLP methods for this task has been hindered by the lack of high-quality publicly available labeled data, largely due to the privacy and regulatory constraints on the use of real patients' information. This paper introduces a new dataset, SDOH-NLI, that is based on publicly available notes and which we release publicly.[1] We formulate SDOH extraction as a natural language inference (NLI) task, and provide binary textual entailment labels obtained from human raters for a cross product of a set of social history snippets as premises and SDOH factors as hypotheses. Our dataset differs from standard NLI benchmarks in that our premises and hypotheses are obtained independently. We evaluate both "off-the-shelf" entailment models as well as models fine-tuned on our data, and highlight the ways in which our dataset appears more challenging than commonly used NLI datasets.

## 1 Introduction

There has been growing recognition that social and behavioral determinants of health (SDOH) play a significant role in shaping health outcomes for individuals and populations. The ability to accurately identify and extract social and behavioral determinants of health from clinical notes can provide valuable insights that can enable healthcare providers to better understand and address the underlying determinants of health that contribute to poor health outcomes and health disparities.

Social determinants of health are frequently recorded in clinical notes as unstructured text, so natural language processing (NLP) can be a valuable tool for extracting actionable insights for care teams. However, research in this area often uses patient records from private health systems' electronic health records (EHRs), which makes it difficult to compare results from other health systems or even replicate the studies. The development and release of high-quality publicly available datasets could enable more reproducible research in this area.

In this work, we introduce a new, public SDOH dataset based on MTSamples.com, an online collection of transcribed medical reports. Our setup is motivated by the use cases of slicing patient populations along social determinant dimensions for population analytics, and retrieving patients with certain social determinants of health to allow for more targeted outreach and intervention. Given a large set of social determinant factors, our goal is to make binary determinations for each patient about whether that patient's notes imply a particular SDOH factor. In other words, for example, we want to be able to find all patients who lack access to transportation, as opposed to just tagging transportation-related spans in their notes, as done in some previous work.

To achieve this goal, we formulate the task as a textual entailment problem, with patient note snippets as the premises, SDOH factors as the hypotheses, and binary entailment labels. We use human annotators to label 1,398 social history snippets according to a curated list of 60 SDOH statements, resulting in a dataset of 29,635 labeled premise-hypothesis examples after some filtering (see Section 3 for details). We release this dataset publicly. We also evaluate state-of-the-art publicly available large language models on our data in a range of different settings (see Section 4).

A notable feature of our entailment dataset is

---

*Corresponding author. lelkes@google.com
†Work completed at Google.
‡Equal leadership.

[1]https://github.com/google-research-datasets/SDOH-NLI

that unlike other entailment datasets, our premises and hypotheses were obtained independently, and we label the full cross product of premises and hypotheses. In traditional entailment datasets, the hypotheses are constructed to be tied to a particular premise; however, in our formulation, the hypotheses are drawn from a large set of SDOH factors that may or may not be discussed in a particular premise (drawn from a clinical note). Since all our text comes from the same domain, we still have a non-negligible fraction of positive entailment labels (albeit with a much larger label imbalance than in standard NLI benchmark datasets).

This requires NLI methods to understand both the premise and the hypothesis, and defeats typical shortcuts that have been observed to work for other entailment datasets, such as guessing the label based on the hypothesis alone, or relying on simple syntactic clues such as the presence of negations (see Section 2.2). Indeed, even though our task does not require domain-specific knowledge, we observe that state-of-the-art models struggle to generalize from common NLI benchmarks to our dataset, and highlight typical failure cases (see Section 4).

We evaluate both off-the-shelf and fine-tuned models in different setups: dual encoders, zero/few-shot prompting, and binary classification. We show that state-of-the-art off-the-shelf models, even if they were fine-tuned on various NLI datasets, do not reliably solve our problem; on the other hand, models fine-tuned on our training set robustly generalize both to unseen notes and to unseen factors, providing evidence for the usefulness of our dataset for model fine-tuning and evaluation.

## 2 Related Work

### 2.1 Social and Behavorial Determinants of Health

There has been a lot of interest in using NLP techniques for SDOH extraction; see Patra et al. (2021) for a recent survey. The range and granularity of SDOH factors vary considerably across different papers. There has also been a range of methods used, from rule-based heuristics, to n-grams, to fine-tuning pretrained Transformer models.

Many previous research studies on SDOH extraction were performed on EHR data from particular health systems and are not released publicly. Exceptions include the i2b2 NLP Smoking Challenge (Uzuner et al., 2008), which classified 502

deidentified medical discharge records for smoking status only, and small number of datasets based on MIMIC-III (Johnson et al., 2016), a large publicly available database of deidentified health records of patients who stayed in critical care units of Beth Israel Deaconess Medical Center between 2001 and 2012. For example, Gehrmann et al. (2018) annotated MIMIC-III with binary labels for certain "phenotypes" including alcohol and substance abuse, and Lybarger et al. (2021) annotated note spans with SDOH information.

We are aware of four other previous SDOH-related papers which used MTSamples data (Wang et al., 2015; Winden et al., 2017; Yetisgen et al., 2016; Yetisgen and Vanderwende, 2017), all of which focused on extracting and tagging SDOH-related spans from social history sections.

Among previous papers, the one methodologically closest to ours is Lituiev et al. (2023) which also formulated SDOH extraction as an entailment problem and evaluated RoBERTa (Liu et al., 2019) fine-tuned on ANLI (Nie et al., 2020) on clinical notes from UCSF, without any in-domain fine-tuning experiments.

### 2.2 Natural Language Inference

Natural language inference (NLI), also called recognizing textual entailment (RTE), has been a very well-studied NLP task; see e.g. Storks et al. (2019); Poliak (2020) for recent surveys. Many standard NLI datasets, such as SNLI (Bowman et al., 2015) or MultiNLI (Williams et al., 2018), are obtained by automatically collecting premises and, for each premise and target entailment label, having human annotators write a hypothesis with the specified entailment relation to the premise. (Even some of the datasets specifically designed to address these datasets' shortcomings, such as ANLI (Nie et al., 2020), follow a similar setup.) It has been observed that this leads to annotation artifacts that lets models do well on these tasks without requiring true understanding of entailment, including by using simple syntactic heuristics (McCoy et al., 2019) or by completely ignoring the premise, and considering only the hypothesis (Gururangan et al., 2018; Poliak et al., 2018). ContractNLI (Koreeda and Manning, 2021) is an example of a previous dataset which used the same fixed set of hypotheses for all the premises.

## 3 Dataset Construction

We scraped all 5,003 medical reports from MTSamples. Within these reports, we obtained 1,030 note sections related to social history by searching for a collection of note section titles identified as synonyms of social history. We then split each note section into sentences, resulting in 3,281 sentences. Many sentences (such as "He is married") appear in multiple notes; after deduplication, we have 1,398 unique sentences.

We manually curated a collection of SDOH factors primarily from two sources, the AHRQ Social Determinants of Health Database (for Healthcare Research and Quality, 2020) and UCSF SIREN's Compendium of medical terminology codes for social risk factors (Arons et al., 2018). We rephrased each factor as a full English sentence stating a fact about a person; e.g. "The person is employed." For binary factors, we included a statement and its negation; e.g. "The person currently drinks alcohol" and "The person currently does **not** drink alcohol." For factors with multiple potential values (such as housing status), we aimed to list all the common options. We grouped these 60 statements into 10 categories: smoking, alcohol use, drug use, employment, housing, food, transportation, health insurance, social support, and financial situation. See the full list in Appendix A.

For each social history sentence, we asked human raters to select all the relevant categories and, within each category, all the statements that are supported by the social history snippet. Each snippet was rated by at least three raters. For each (snippet, statement) pair, we took the majority vote of raters to get binary entailment labels. Rater agreement was high, with a Krippendorff's alpha of 0.97 (computed over the binary entailment labels provided by different raters for (premise, hypothesis) pairs).

We removed all SDOH statements which were not entailed by any of the social history snippets as well as all note sentences that were not relevant to any SDOH categories. Finally, after inspecting the dataset, we removed three SDOH factors ("The person is stably housed.", "The person has social support.", "The person does not have social support.") because raters weren't able to consistently give correct ratings. That left us with 787 unique sentences and 38 SDOH factors. Since each snippet would typically only entail one factor or a small number of factors, the resulting dataset is heavily imbalanced: only $4.6\%$ of the labels are positive.

We split the dataset along the snippets into training, validation, and test sets with a 70:15:15 ratio, with the following modification: we remove a single pair of factors, "The person lives alone" and "The person does not live alone," from the training and validation sets, in order to evaluate fine-tuned models' ability to generalize to unseen factors. In other words, the training, validation, and test sets have disjoint note snippets but the same SDOH factors, except for the test set which also contains an additional two factors that are not present in the training or validation sets.

## 4 Model Evaluation

Since the typical use case is retrieving patients with a particular SDOH factor, and because of the heavy label imbalance in our dataset, we use precision, recall, and F1 score as our evaluation metrics.

We evaluate state-of-the-art public models in four different setups:

- Treating the problem as a retrieval task with SDOH factors as queries and note snippets as documents. We evaluate Sentence-T5 (Ni et al., 2022a) and GTR (Ni et al., 2022b), two state-of-the-art dual encoder models. We select the cosine similarity threshold which maximizes F1 score on the training set.

- A general-purpose NLI model. We evaluate the SENTLI model (Schuster et al., 2022), a state-of-the-art NLI model (T5 large fine-tuned on the SNLI (Bowman et al., 2015), MNLI (Williams et al., 2018), ANLI (Nie et al., 2020), Fever (Thorne et al., 2018), and VitaminC (Schuster et al., 2021) datasets). To help the model adapt to the label imbalance in our dataset, we also evaluate it with re-tuning the threshold for positive prediction to maximize F1 score on our training set.

- Zero/few-shot prompting. We evaluate Flan-T5 XXL (Chung et al., 2022) and Flan-UL2 (Tay et al., 2023) in both a zero- and a few-shot setting. These models instruction tuned for a large set of tasks, including NLI tasks. (See Appendix B for details.)

- Fine-tuning experiments. We fine-tune T5 and Flan-T5 on our dataset (SDOH-NLI), on ANLI, and on a mixture of both at various model sizes. (See Appendix C for details.)

| Model | Precision | Recall | F1 |
|---|---|---|---|
| Sentence-T5 11B | .2826 | .523 | .3669 |
| GTR XXL | .3148 | .4885 | .3829 |
| SENTLI | .4438 | .8161 | .5749 |
| SENTLI, threshold tuned | .5721 | .7299 | .6414 |
| Flan-T5 XXL 0-shot | .6867 | .6552 | .6706 |
| Flan-T5 XXL 5-shot | .6297 | .6839 | .6556 |
| Flan-UL2 0-shot | .5922 | .7011 | .6421 |
| Flan-UL2 5-shot | .5536 | .7126 | .6231 |
| Flan-T5 XXL finetuned on ANLI | .356 | **.8736** | .5058 |
| Flan-T5 XXL finetuned on SDOH-NLI | **.9295** | .8333 | **.8788** |
| Flan-T5 XXL finetuned on ANLI + SDOH-NLI | .8765 | .8563 | .8663 |

Table 1: Scores of retrieval-based, general-purpose NLI, in-context, and fine-tuned models on the SDOH test set. See Appendix C for additional fine-tuning experiments.

## 4.1 Results

Table 1 shows selected results (see Appendix D for more model fine-tuning metrics). First, we observe that while our problem can naturally be framed as an information retrieval problem, even state-of-the-art retrieval models perform poorly on it; formulating the problem as natural language inference yields dramatically better results.

However, even powerful models fine-tuned for NLI, either on NLI datasets alone or as part of the much larger Flan collection, do not reliably solve our problem, with an F1 score of at most .67 on the test set. Even T5 small fine-tuned on our dataset outperforms the largest off-the-shelf models, highlighting the added value of our dataset.

When prompting instruction-tuned models, including few-shot examples does not appear to add any value compared to a zero-shot setup. We conjecture that this is because the models are already familiar with the NLI task and the prompt format from instruction tuning, and presenting them with a small number of additional examples is not sufficient for teaching them the difference between this task and the NLI benchmark datasets they have seen during fine-tuning.

For fine-tuned models below XXL size, when fine-tuning on the SDOH-NLI training set only, we observed poor generalization to the held-out SDOH factors in the test set (see Appendix D for detailed metrics). Because of this, why we experimented with fine-tuning on a combination of our data and ANLI R1, a challenging general-purpose NLI dataset of a similar size. For smaller models, fine-tuning on this mixture enabled robust generalization to unseen factors without sacrificing overall test performance.

## 4.2 Discussion

What makes our dataset challenging for models fine-tuned on standard NLI datasets? We emphasize that it is not that it requires any specialized (e.g., medical) knowledge: most of our examples describe everyday situations in plain English (e.g., "The patient is retired on disability due to her knee replacements"); human raters without any domain-specific training had no difficulty understanding them. We conjecture that the main difference is that the hypotheses of typical NLI datasets are written to satisfy a given entailment label for a given premise, whereas ours are obtained independently from the premises, and therefore their entailment status can be more subtle and ambiguous. These models also struggled with distinguishing between statements about the present and the past; e.g., Flan-UL2 erroneously predicting that "He is not a smoker" entails "The person wasn't a smoker in the past." Also, our dataset contains a lot of irrelevant hypotheses and requires the model to correctly classify all of those; we see off-the-shelf models occasionally giving them positive labels (e.g. predicting that "She was living alone and is now living in assisted living." implies "The person drank alcohol in the past."), hurting precision. As an example, Flan-T5 fine-tuned on ANLI had comparable recall to our best models, but much worse precision.

## 5 Conclusion

In this work, we introduced SDOH-NLI, a new entailment dataset containing social history snippets as premises and social and behavioral determinants of health as hypotheses. Our dataset was designed both to reflect realistic use cases of SDOH extraction in clinical settings as well as to provide a high quality entailment dataset to support broader NLI research. We evaluated baseline methods using state-of-the-art public models in a variety of setups, and highlighted novel and challenging features of our dataset.

## Acknowledgments

The authors would like to thank Jonas B. Kemp, Von Nguyen, Birju Patel, Martin G. Seneviratne, Andy Strunk, and Vinh Q. Tran for their valuable feedback and discussions.

## Limitations

Our dataset is English-only, and reflects the American healthcare system. While a lot of the social and behavioral determinants of health mentioned in the data could apply elsewhere, too, their distribution and the language used to describe them could reflect U.S. norms. Also, since the dataset is based on transcription samples, the text can be cleaner than in some other settings (such as notes in EHRs), where the task could be more challenging due to typos and abbreviations that do not appear in our dataset. In such settings, performance could be improved by first using the methods of Rajkomar et al. (2022) to decode abbreviations, but we have not included this in our evaluations. Finally, our dataset only contains short note snippets, and we have not evaluated the models' ability to reconcile contradictory statements or reason about the chronology of information in longer patient records. For longer contexts, especially if the social history sections don't fit in the Transformer model's context window, we recommend evaluating the methods of Schuster et al. (2022).

## Ethics Statement

Machine learning research in the clinical domain has many ethical considerations. Given the nature of this work is to identify opportunities for care teams to improve the health outcomes of patients due to factors commonly not addressed, we think it contributes to human well-being and avoids harms.

The use of public, non-identifiable data that is released for the NLP community helps balance the need to have reproducible (i.e., honest and trustworthy) data to enable technical advances while limiting the need for sensitive, private medical records for research. We acknowledge that the purpose of the work is to identify SDOH to provide additional help and services to patients, and we warn against any use to deny patients care or services based on any factor that can be identified. Although we picked a large number of SDOH factors to test our method, we acknowledge that there may be additional factors that are important for specific patients and populations, so we encourage researchers to reflect on those possible factors and create datasets to help others study them, as well.

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

# A   Labeling Setup

Raters were given the following instructions:

"In this task, you will be given a list of snippets from transcribed medical records, describing a person's social history. Your job is to select the categories that are relevant to the snippet and, in each category, select statements about the person that are supported by the snippet.

Only select categories and statements relevant to the subject of the sentence, NOT if they apply to someone else (such as the subject's relatives).

The source of the medical transcript samples is a public dataset (MTSamples.com), and not your, other raters' or users health transcript data."

For each snippet, they were required select one or more of the following categories (including "None of the above" if none of the categories were relevant). If they selected a category, the statements within the category would be displayed, and they were required to select one of statements within that category, or "None of the above":

- Smoking

    - The person is currently a smoker.
    - The person is currently not a smoker.
    - The person was a smoker in the past.
    - The person wasn't a smoker in the past.
    - None of the above

- Alcohol use

    - The person currently drinks alcohol.
    - The person currently does not drink alcohol.
    - The person drank alcohol in the past.
    - The person did not drink alcohol in the past.
    - None of the above

- Drug use

    - The person is a drug user.
    - The person is not a drug user.
    - The person was a drug user in the past.
    - The person wasn't a drug user in the past.
    - The person uses cocaine.
    - The person does not use cocaine.
    - The person used cocaine in the past.
    - The person did not use cocaine in the past.
    - The person uses marijuana.

– The person does not use marijuana.

– The person used marijuana in the past.

– The person did not use marijuana in the past.

– The person uses opioids (e.g., heroin, fentanyl, oxycodone).

– The person does not opioids (e.g., heroin, fentanyl, oxycodone).

– The person used opioids in the past (e.g., heroin, fentanyl, oxycodone).

– The person did not use opioids in the past (e.g., heroin, fentanyl, oxycodone).

– None of the above

• Employment

– The person is employed.

– The person is not employed.

– The person is employed part time.

– The person is a student.

– The person is a homemaker.

– The person is retired due to age or preference.

– The person is retired due to disability.

– The person is retired due to an unknown reason.

– None of the above

• Housing

– The person lives in their own or their family's home.

– The person lives in a housing facility.

– The person is homeless.

– The person is stably housed.

– The person's housing is unsuited to their needs.

– None of the above

• Food

– The person is able to obtain food on a consistent basis.

– The person is not able to obtain food on a consistent basis.

– The person has consistent fruit and vegetable intake.

– The person does not have consistent fruit and vegetable intake.

– None of the above

• Transportation

– The person has access to transportation.

– The person does not have access to transportation.

– The person has access to a car.

– The person does not have access to a car.

– The person has access to public transit.

– The person does not have access to public transit.

– The person has issues with finding transportation.

– None of the above

• Health insurance

– The person has private health insurance.

– The person is on Medicare.

– The person is on Medicaid.

– The person does not have health insurance.

– None of the above

• Social support

– The person has social support.

– The person does not have social support.

– The person lives alone.

– The person does not live alone.

– None of the above

• Financial situation

– The person is below the poverty line.

– The person is above the poverty line.

– The person is able to afford medications.

– The person is not able to afford medications.

– None of the above

• None of the above

## B  Prompting Setup

Our goal with the design of our zero/few-shot experiments was to stay close to how the models were trained and evaluated on similar NLI tasks. Since the models we used were fine-tuned on the Flan collection (Longpre et al., 2023), which includes several NLI datasets, we reused some of the Flan collections's prompt templates.

In particular, we used the templates for the SNLI and MNLI dataset which were the most relevant to our data. (We excluded templates for other datasets such as RTE, ANLI, or WNLI because the wording

of some of the prompts could be slightly misleading; e.g. by referring to the premise as a "paragraph.") For each example, we chose a prompt template uniformly at random from the 19 templates. For few-shot experiments, we first picked a random positive example from the training set, then picked the other few-shot examples uniformly at random from the training examples. (Without explicitly forcing at least one example to be positive, it would be very likely that all the few-shot examples would be negative, given the label imbalance in the dataset. We tried dropping that requirement and sampling all examples at random, which resulted in a slight drop in model performance, as we expected.)

Since SNLI and MNLI have three answer options ("yes," "it is not possible to tell," "no"), we kept all three of these options in the prompt template, even though our dataset is binary. We experimented with dropping either the "no" or the "it is not possible to tell" option; both of these deviations from the original prompts resulted in slight decreases in model performance.

We use rank classification to obtain binary labels (as is customary): i.e., instead of decoding a model prediction, we score the three answer options (for three copies of each input example) and consider the prediction positive if "yes" has the highest score. Similarly to our experiment with SENTLI, we also tried taking softmax over the three options and using a fixed threshold for "yes" which maximizes F1 score on the training set; this resulted in a small drop in test F1 score (e.g., .6501 to .6222 for Flan-UL2).

## C Fine-tuning Setup

All models were fine-tuned using the T5X framework (Roberts et al., 2022) on TPUv3 chips for 10k steps with batch size 32 and learning rate 1e-4, with the exception of the T5 small size models which were finetuned for 50k steps. For each model, we picked the checkpoint with the highest F1 score on the validation set.

## D Additional Model Fine-tuning Results

See Table 2 for the full set of results from our fine-tuning experiments. Here we also include metrics on the subset of the test set consisiting on the new SDOH factors not included in the training and validation sets to highlight the differences in models' ability to generalize to unseen factors.

Note that since Flan-T5 models between sizes large and XL fine-tuned on either ANLI or our SDOH dataset alone underperformed the same models fine-tuned on the combined dataset, we did not perform these dataset ablations on smaller Flan-T5 models or on T5 below the XXL size.

| Model | Size | Dataset | Test P | Test R | Test F1 | New factors P | New factors R | New factors F1 |
|-------|------|---------|--------|--------|---------|---------------|---------------|----------------|
| T5 | small | ANLI + SDOH-NLI | .7987 | .7069 | .7500 | .6667 | .0833 | .1481 |
| T5 | base | ANLI + SDOH-NLI | .7396 | .8161 | .7760 | .75 | .375 | .5 |
| T5 | large | ANLI + SDOH-NLI | .8247 | .7299 | .7744 | .75 | .125 | .2143 |
| T5 | XL | ANLI + SDOH-NLI | .8391 | .8391 | .8391 | .9231 | .5 | .6486 |
| T5 | XXL | ANLI | .8391 | .8391 | .8391 | .75 | .1 | .8571 |
| T5 | XXL | SDOH-NLI | .8859 | .7586 | .8173 | .8571 | .25 | .3871 |
| T5 | XXL | ANLI + SDOH-NLI | .9024 | .8506 | .8757 | .8889 | 1. | .9412 |
| Flan-T5 | small | ANLI + SDOH-NLI | .8075 | .7471 | .7761 | .6667 | .0833 | .1481 |
| Flan-T5 | base | ANLI + SDOH-NLI | .7956 | .8276 | .8113 | .6216 | .9583 | .7541 |
| Flan-T5 | large | ANLI | .1777 | .8506 | .2939 | .2421 | .9583 | .3866 |
| Flan-T5 | large | SDOH-NLI | .8041 | .6839 | .7391 | .2308 | .125 | .1622 |
| Flan-T5 | large | ANLI + SDOH-NLI | .8212 | .8448 | .8329 | .8846 | .9583 | .92 |
| Flan-T5 | XL | ANLI | .2868 | .8851 | .4332 | .7667 | .9583 | .8519 |
| Flan-T5 | XL | SDOH | .9071 | .7299 | .8089 | 1. | .125 | .2222 |
| Flan-T5 | XL | ANLI + SDOH-NLI | .8647 | .8448 | .8547 | .8846 | .9583 | .92 |
| Flan-T5 | XXL | ANLI | .3560 | .8736 | .5058 | .7931 | .9583 | .8679 |
| Flan-T5 | XXL | SDOH-NLI | .9295 | .8333 | .8788 | 1. | .875 | .9333 |
| Flan-T5 | XXL | ANLI + SDOH-NLI | .8765 | .8563 | .8663 | .8846 | .9583 | .92 |

Table 2: Precision, recall, and F1 scores of various models on the full test set and on the subset of the test set consisting of the two held-out factors.