# OpenReview forum: "SDOH-NLI: a Dataset for Inferring Social Determinants of Health from Clinical Notes"
_EMNLP/2023/Conference — EMNLP 2023 Findings_

### Official Review · Reviewer_Yo1y · 2023-08-01

**Soundness:** 3

**Excitement:**

3: Ambivalent: It has merits (e.g., it reports state-of-the-art results, the idea is nice), but there are key weaknesses (e.g., it describes incremental work), and it can significantly benefit from another round of revision. However, I won't object to accepting it if my co-reviewers champion it.

**Paper Topic And Main Contributions:**

This paper is about the SDOH-NLI dataset, which is a new dataset that can help healthcare providers identify opportunities to provide appropriate care and address disparities. The paper addresses the problem of identifying social and behavioral determinants of health (SDOH) from clinical notes, which can be challenging due to the unstructured nature of these notes. The main contribution of the paper is the creation of the SDOH-NLI dataset, which is based on publicly available notes and is formulated as a natural language inference task. The paper also evaluates several state-of-the-art models on this dataset and provides insights into the performance of these models.

**Reasons To Accept:**

The strengths of this paper include the creation of a new dataset, SDOH-NLI, which can help healthcare providers identify opportunities to provide appropriate care and address disparities. The dataset is based on publicly available notes and is formulated as a natural language inference task, which makes it a valuable resource for researchers working on natural language processing techniques for healthcare applications. The paper also evaluates several state-of-the-art models on this dataset and provides insights into the performance of these models, which can help guide future research in this area.

**Reasons To Reject:**

The experiments are not enough.

**Reproducibility:**

4: Could mostly reproduce the results, but there may be some variation because of sample variance or minor variations in their interpretation of the protocol or method.

**Reviewer Confidence:**

3: Pretty sure, but there's a chance I missed something. Although I have a good feel for this area in general, I did not carefully check the paper's details, e.g., the math, experimental design, or novelty.

---

> ### Author Rebuttal · Authors · 2023-08-28
>
> We thank the reviewer for their review. Unfortunately we are not quite sure what the reviewer meant by the statement that “The experiments are not enough” or why they gave us a low rating on soundness; we would have appreciated more specific feedback. Our paper reports experiments in four different setups (zero-shot information retrieval, off-the-shelf NLI model, zero and few-shot prompting, and fine-tuning). Due to the page limit for a short paper, we weren’t able to report all our experimental results in the main paper, but Appendix D shows additional experimental results on 18 different fine-tuned models. If there are any specific experiments that the reviewer would like us to include, we'll be happy to consider reporting them in the paper.
>
> We would also like to address the reviewer’s reproducibility rating. We take pride in publicly releasing our full dataset; it was included in our submission. Also, we exclusively used publicly available, open-source models, which should make it possible for anyone to reproduce and verify our results. For the zero/few-shot prompting experiments, we gave a detailed description of our setup in Appendix B. For the fine-tuning experiments, all relevant hyperparameters are listed in Appendix C. If there are any details still missing for the reviewer to be able to reproduce our results, we would be very happy to add them to the paper.

---

### Official Review · Reviewer_sNnf · 2023-08-05

**Soundness:** 3

**Excitement:**

3: Ambivalent: It has merits (e.g., it reports state-of-the-art results, the idea is nice), but there are key weaknesses (e.g., it describes incremental work), and it can significantly benefit from another round of revision. However, I won't object to accepting it if my co-reviewers champion it.

**Paper Topic And Main Contributions:**

The paper presents a new dataset, SDOH-NLI, which is designed to enhance the extraction of Social and Behavioral Determinants of Health (SDOH) from clinical notes using Natural Language Processing (NLP). It reformulates SDOH extraction as a natural language inference task, with binary textual entailment labels assigned by human raters. The dataset, which is based on clinical notes, is also made publicly available.

**Questions For The Authors:**

The reliance on human raters for assigning binary textual entailment labels might introduce bias or errors into the dataset, affecting its quality. Who are the human annotators? Are they experts?

**Reasons To Accept:**

The paper tackles the vital issue of extracting SDOH from clinical notes, which is a significant step towards identifying opportunities for proper care and addressing disparities in healthcare.

It introduces a new dataset, SDOH-NLI, overcoming the hurdle of the lack of high-quality publicly available labeled data. This could accelerate the progress of NLP methods in this field.

The authors' innovative approach of formulating SDOH extraction as a natural language inference task could open new ways of dealing with such problems.

They provide an assessment of both off-the-shelf entailment models and models fine-tuned on their data, offering a comprehensive evaluation.

**Reasons To Reject:**

The paper doesn't elaborate on how privacy and regulatory constraints were navigated in the creation of this public dataset, which might raise some ethical and legal concerns.

Being a subjective domain, the Inter annotator agreement is expected to be low. Rater agreement was high, with a Krippendorff’s alpha of 0.97, seems to be unrealistic.

This work need a lot more details about the annotation guidelines, the data collection and the description on the nature of the dataset. Ad detailed description on how is it useful for stakeholders and what measures are taken for explanation and trustworthiness of model development is required.

**Reproducibility:**

5: Could easily reproduce the results.

**Reviewer Confidence:**

5: Positive that my evaluation is correct. I read the paper very carefully and I am very familiar with related work.

---

> ### Author Rebuttal · Authors · 2023-08-28
>
> We thank the reviewer for their review and for appreciating the importance of the problem and the effect our dataset could have in accelerating progress in the field. Let us try to address the reviewer’s critiques and questions.
>
> ___
>
> > The paper doesn't elaborate on how privacy and regulatory constraints were navigated in the creation of this public dataset, which might raise some ethical and legal concerns.
>
> We tried to briefly address this in the introduction and in our ethics statement. As we pointed out, a lot of previous research in the area relied on protected health information (PHI) from the electronic health records of particular health systems, preventing researchers from sharing their data publicly and making their results non-reproducible. By using already **fully public, non-PHI medical transcription samples**, we are able to share our dataset with the research community without jeopardizing patient privacy. (All of the text we used in our research is already publicly available on MTSamples.com.) The creation, usage, and planned release of this data set was approved by a rigorous cross-functional review process at our institution.
>
> ___
>
> > Being a subjective domain, the Inter annotator agreement is expected to be low. Rater agreement was high, with a Krippendorff’s alpha of 0.97, seems to be unrealistic.
>
> We believe that raters overall did a great job at correctly labeling our examples. We carefully examined the ratings; as we mentioned in Section 3, after careful inspection, we dropped three SDOH factors from our datasets that appeared too ambiguous and for which we weren’t satisfied with the quality of the ratings. Note also that Krippendorff’s alpha was computed over the binary entailment labels provided by different raters for (premise, hypothesis) pairs. The label imbalance in our dataset (i.e., that the vast majority of premises very clearly did not entail the corresponding hypotheses) also contributed to the high alpha.
>
> In case it is helpful, here is a more detailed breakdown of rater agreement, grouped by social determinant category. For each category, the listed number is the proportion of snippets for which all three raters fully agree on the ratings for *all* the statements in that category.
>
> | Category            | Rater Agreement |
> |---------------------|-----------------|
> | Food                | 99.42%          |
> | Health insurance    | 99.02%          |
> | Financial situation | 98.56%          |
> | Transportation      | 97.47%           |
> | Alcohol use         | 90.11%           |
> | Smoking             | 89.37%           |
> | Employment          | 88.76%           |
> | Drug use            | 88.31%           |
> | Housing             | 87.85%           |
> | Social support      | 74.5%           |
>
> Note that these metrics were computed *before* we dropped a few ambiguous factors from the two categories with the lowest rater agreement (housing and social support).
>
> ___
>
> > This work need a lot more details about the annotation guidelines, the data collection and the description on the nature of the dataset. As detailed description on how is it useful for stakeholders and what measures are taken for explanation and trustworthiness of model development is required.
>
> We agree with the reviewer that there could be more discussion of these topics; the page limit for short papers imposes tight constraints on how much detail we can go into on any particular question. For the annotation guidelines, please see Appendix A which has the full instructions given to raters.
>
> ___
>
> > The reliance on human raters for assigning binary textual entailment labels might introduce bias or errors into the dataset, affecting its quality. Who are the human annotators? Are they experts?
>
> We have carefully inspected the dataset to make sure the labels are high quality. All of our raters were proficient in English but they were not required to be experts in the area. As we mentioned in Section 4.2, most of our examples describe everyday situations in plain English (e.g.,"The patient is retired on disability due to her knee replacements"), and our human raters without any domain-specific training had no difficulty understanding them.

---

### Official Review · Reviewer_YiJH · 2023-08-08

**Soundness:** 3

**Excitement:**

3: Ambivalent: It has merits (e.g., it reports state-of-the-art results, the idea is nice), but there are key weaknesses (e.g., it describes incremental work), and it can significantly benefit from another round of revision. However, I won't object to accepting it if my co-reviewers champion it.

**Paper Topic And Main Contributions:**

The paper is a resource paper that introduces a new dataset, SDOH-NLI, for the task of recognizing social and behavioral determinants of health (SDOH) from clinical notes. The authors frame the extraction of SDOH characteristics as an NLI task, where snippets of social history are premises and SDOH characteristics are hypotheses. The authors claim the dataset is unique and unlike other NLI datasets since premises and hypotheses were collected independently one of another. The paper also tests the performance of baseline models on the newly developed dataset, including retrieval models and NLI models, in both zero-shot and fine-tuned settings. The results show that the dataset is challenging in case when NLI models are fine-tuned on existing NLI datasets, showing a big change in performance as opposed to fine-tuning on the new dataset.


**Questions For The Authors:**

A: In line 89, what is the relation between the fact that there is a non-negligible fraction of positive entailment labels and coming from the same domain?
B: In line 226, it is mentioned that the pair of factors ("The person lives alone" / "The person does not live alone") was removed from the training and validation sets. How did the models perform on this factor?
C: Was the F1 score macro-averaged? Do you maybe have a confusion matrix or results showing the difference between positive and negative class performance, considering the heavy imbalance?
D: In line 323, it is said that the dataset contains a lot of irrelevant hypotheses. Why was the dataset constructed in this way if it led to irrelevant hypotheses and noise?

**Reasons To Accept:**

The paper addresses an important issue of detection of social and behavioral determinants of health from clinical notes, which can improve opportunities of patients and access to appropriate healthcare options.
The paper introduces a novel dataset, with lots of manual annotation effort, that could be useful for the NLP community and NLI research.
The dataset is evaluated with various models, in both zero-shot and fine-tuned settings, and the results are reported.
The paper is well-written and easy to follow.


**Reasons To Reject:**

The usability of the dataset is questionable. Since it was constructed as a cross product of 787 unique sentences and 38 SDOH factors, this yielded around 30 thousand instances, where the same premise (snippet) is always repeated 38 times and the same hypothesis always repeated 787 times. This makes the dataset not just heavily imbalanced (only 4% positive examples) but also heavily redundant with this huge amount of repeating sentences. This can lead the models to pick up statistical patterns and not really rely on deeper understanding.
Furthermore, most of the snippets of social history are literally snippets and not even formed to full sentences (such as "no alcohol use", "unemployed", "1-2 ppd Cigarettes"). It is not completely surprising that models fine-tuned on a dataset like ANLI, that has a whole paragraph as a premise and a long sentence as a hypothesis, would struggle on this dataset. The premise "unemployed" does not necessarily contradict "This person is employed" since the subject is not even specified. It is not clear how useful would an NLI model fine-tuned on SDOH-NLI be for entailing SDOHs from unseen clinical notes considering the data, and would it be better to frame the problem as pure information extraction or information retrieval (in a form of similarity search).
The authors observe that the problem can naturally be framed as an information retrieval. This is true in the case when the whole clinical notes document, or a corpus of these documents, is used to determine SDOHs from a given query. But how exactly were these retrieval models fine-tuned for this specific dataset? Retrieval models are not meant to work on pairs of two short sentences at a time.

**Reproducibility:**

3: Could reproduce the results with some difficulty. The settings of parameters are underspecified or subjectively determined; the training/evaluation data are not widely available.

**Reviewer Confidence:**

4: Quite sure. I tried to check the important points carefully. It's unlikely, though conceivable, that I missed something that should affect my ratings.

---

> ### Author Rebuttal · Authors · 2023-08-28
>
> We thank the reviewer for the careful review, and thank the reviewer for appreciating the importance of the problem and the effort that went into curating the dataset.
>
> We also thank the reviewer for the detailed critique of our datasets. We respectfully disagree that these features of our dataset would be reasons to reject the paper; we like to think of these as unique features that make the dataset novel and which represent real-world use cases. We will try to address these critiques below.
>
> ___
>
> > This makes the dataset not just heavily imbalanced (only 4% positive examples) but also heavily redundant with this huge amount of repeating sentences.
>
> We would argue that label imbalance makes the task **more challenging** and also more reflective of the motivating application. Real-world data is even more heavily imbalanced: electronic health records can contain many notes for a given patient, and individual notes are often long and address multiple topics. The total amount of text for even a single patient can be orders of magnitude more than what fits in a typical Transformer model’s context window. Inferring which patients have a particular social determinant of health requires processing all that text, most of which will be irrelevant.
>
> We would argue that this setup is also a **better test of how well models recognize neutral entailment relationships**. For standard NLI benchmark datasets, hypotheses are typically at least somewhat relevant to the corresponding premises. Our dataset tests whether models robustly generalize to cases when hypotheses and premises might be completely unrelated.
>
> ___
>
> > This can lead the models to pick up statistical patterns and not really rely on deeper understanding.
>
> As we briefly mention in Section 2.2, that models can pick up statistical patterns has actually been observed for NLI datasets where premises and hypotheses are *not* collected independently. When hypotheses are written specifically for each premise, there are often simple syntactic artifacts (such as negation) that let the model guess the entailment relationship without any real understanding. In fact, it has been observed that models trained on hypotheses only in such datasets, completely ignoring the premises, achieve decent performance [1,2]. These shortcuts become impossible when, as in our dataset, hypotheses and premises are obtained independently and the model has to correctly label the entire cross product. As an example, Flan-T5 XXL fine-tuned on hypotheses only in SDOH-NLI (with keeping the rest of the setup and hyperparameters the same as for our best-performing model) learns the constant 0 classifier, getting an F1 score of 0. (It achieves an AUC-PR of .2, compared to .95 for Flan-T5 XXL fine-tuned on both hypotheses and premises.)
>
> ___
>
> > most of the snippets of social history are literally snippets and not even formed to full sentences (such as "no alcohol use", "unemployed", "1-2 ppd Cigarettes").
>
> Again, we would argue that this makes our dataset **more realistic**. (If anything, the text in MTSample is cleaner than in many real-world EHRs.) If, as you write, models fine-tuned on standard NLI datasets such as ANLI are to be expected to struggle in such real-world scenarios, that seems to be an argument for introducing a new dataset that captures the messiness of real-world text!
>
> ___
>
> To answer the questions:
>
> > A: In line 89, what is the relation between the fact that there is a non-negligible fraction of positive entailment labels and coming from the same domain?
>
> We meant to make the point that although labeling the full cross product of independently collected premises and hypotheses has desirable properties, it wouldn’t work if the sentences were e.g. just randomly sampled from some large diverse text corpus (since most sentence pairs would be unrelated to each other). Our dataset has enough diversity to make the task challenging, but not too much that the fraction of positive entailment labels would become negligible.
>
> ___
> > B: In line 226, it is mentioned that the pair of factors ("The person lives alone" / "The person does not live alone") was removed from the training and validation sets. How did the models perform on this factor?
>
> Please see the last three columns in Table 2, Appendix D; they show the fine-tuned models’ performance on these factors.
>
> ___
>
> > C: Was the F1 score macro-averaged? Do you maybe have a confusion matrix or results showing the difference between positive and negative class performance, considering the heavy imbalance?
>
> No. All the precision, recall, and F1 scores are for the positive class. Here is the confusion matrix on the test set for Flan-T5 XXL fine-tuned on SDOH-NLI:
>
>
> | | |
> |:---|---:|
> |3919|11|
> |29|145|
>
>
> ___
>
> > D: In line 323, it is said that the dataset contains a lot of irrelevant hypotheses. Why was the dataset constructed in this way if it led to irrelevant hypotheses and noise?
>
> We don’t view hypothesis-premise pairs with neutral relation as noise but as a way to improve the robustness of the model in distinguishing between neutral and non-neutral relations. A similar approach was also followed in ContractNLI [3]. Including both more closely related pairs and more distant pairs exposes the model to a more diverse set of pairs (both in training and evaluation).
>
> ___
>
> References:
>
> [1] Gururangan, Suchin, et al. "Annotation Artifacts in Natural Language Inference Data." Proceedings of the 2018 Conference of the North American Chapter of the Association for Computational Linguistics: Human Language Technologies, Volume 2 (Short Papers). 2018.
>
> [2] Poliak, Adam, et al. "Hypothesis Only Baselines in Natural Language Inference." Proceedings of the Seventh Joint Conference on Lexical and Computational Semantics. 2018.
>
> [3] Koreeda, Yuta, and Christopher D. Manning. "ContractNLI: A Dataset for Document-level Natural Language Inference for Contracts." Findings of the Association for Computational Linguistics: EMNLP 2021. 2021.

---

### Meta-Review · Area_Chair_1PFg · 2023-09-19

**Recommendation:** 3

**Metareview:**

The paper introduces the SDOH-NLI dataset for recognizing social and behavioral determinants of health (SDOH) from clinical notes. The authors frame the problem as a natural language inference (NLI) task and also provide a baseline performance using multiple models. Reviews point to the relevance and novelty of the work, but also highlight several key concerns such as data imbalance, redundancy, and issues related to the study's design and methodology. All reviewers have provided a "Good" soundness score but are "Ambivalent" regarding their excitement. All reviewers agree on the paper's relevance in addressing an important issue in healthcare, specifically the extraction of SDOH factors from clinical notes. This could potentially aid in improving healthcare equity and decision-making. Reviewers 2 and 3 indicate a lack of detail about the annotation guidelines, data collection, and the utility of the dataset to stakeholders. Reviewer 2 expresses concern about the lack of discussion on navigating privacy and regulatory constraints in data collection, which is crucial given the sensitive nature of healthcare data. Given the dataset's potential value and the paper's overall soundness, the paper is recommended to be accepted into Findings. Authors are encouraged to address the listed concerns to enhance the paper's quality and impact.

---

### Decision · Program_Chairs · 2023-10-07

**Decision:**

Accept-Findings

**Comment:**

The paper introduces the SDOH-NLI dataset for recognizing social and behavioral determinants of health (SDOH) from clinical notes. The authors frame the problem as a natural language inference (NLI) task and also provide a baseline performance using multiple models. Reviews point to the relevance and novelty of the work, but also highlight several key concerns such as data imbalance, redundancy, and issues related to the study's design and methodology. All reviewers have provided a "Good" soundness score but are "Ambivalent" regarding their excitement. All reviewers agree on the paper's relevance in addressing an important issue in healthcare, specifically the extraction of SDOH factors from clinical notes. This could potentially aid in improving healthcare equity and decision-making. Reviewers 2 and 3 indicate a lack of detail about the annotation guidelines, data collection, and the utility of the dataset to stakeholders. Reviewer 2 expresses concern about the lack of discussion on navigating privacy and regulatory constraints in data collection, which is crucial given the sensitive nature of healthcare data. Given the dataset's potential value and the paper's overall soundness, the paper is recommended to be accepted into Findings. Authors are encouraged to address the listed concerns to enhance the paper's quality and impact.